# Multiple Instance Learning and Local Attention for Ischemic Stroke Infarct Segmentation on Early Acute Baseline CTA

**Mahsa Mojtahedi** [1]                                                    S.M.MOJTAHEDI@AMSTERDAMUMC.NL

**Riaan Zoetmulder**[1]                                                    R.ZOETMULDER@AMSTERDAMUMC.NL

**Mart van Blokland**[2]                                                   MVANBLOKLAND@NICO-LAB.COM

**Renan Sales Barros**[2]                                                  RSBARROS@NICO-LAB.COM

**Miou Koopman**[1]                                                        M.S.KOOPMAN@AMSTERDAMUMC.NL

**Charles Majoie**[1]                                                      C.B.MAJOIE@AMSTERDAMUMC.NL

**Efstratios Gavves**[3]                                                   EGAVVES@UVA.NL

**Bart Emmer**[1]                                                          B.J.EMMER@AMSTERDAMUMC.NL

**Henk Marquering**[1]                                                     H.A.MARQUERING@AMSTERDAMUMC.NL

[1] *Department of Radiology and Nuclear Medicine, Amsterdam UMC, Amsterdam, The Netherlands*

[2] *Nico.lab, Amsterdam, The Netherlands*

[3] *University of Amsterdam, Amsterdam, The Netherlands*

**Editors:** Under Review for MIDL 2021

## Abstract

CT is the most available imaging modality for assessing the condition of acute ischemic stroke patients. However, early signs of infarction are barely visible on CT scans, and expert infarct assessments on these images are highly variable. We investigate whether weak labels in conjunction with Multiple-Instance Learning (MIL) can be used to segment early signs of infarction on baseline CTA scans. We propose an alternative to the standard MIL attention, local attention, which exploits the symmetry of the brain. We compare our infarct volume predictions with 3 clinically validated CTP software packages and observe that unlike standard attention, local attention's performance is comparable with the CTP software. This MIL approach including local attention therefore allows for infarct assessment on CTA without implementing the burdensome and time-demanding CTP scans.

**Keywords:** CT Angiography, Stroke, Infarct, Multiple Instance Learning, Attention

## 1. Introduction

The extent of the ischemic core is a major determinant in the evaluation for stroke patient triage and can be determined by brain imaging (Campbell and Khatri, 2020). The main imaging modality used for acute stroke patients is Computed Tomography (CT). However, in the early acute phase of stroke the attenuation change of the infarcted tissue on CT scans is minimal to the extent that experts' opinion on the infarct is highly variable (van Horn et al., 2021). Alternatively, multiple clinically validated software packages can infer the infarct core from CT perfusion (CTP) scans. However, unlike CT Angiography (CTA) and non-contrast CT(NCCT), CTP scans are not commonly performed in many hospitals. Therefore, automatic infarct core prediction on CTA scans would be valuable for stroke diagnosis and triage. The two main challenges of this task are minimal visibility of the lesion and absence of accurate fully annotated infarct labels for supervised learning. We aim to explore whether a weakly-supervised multiple instance learning (MIL) approach allows for infarct core prediction via key instance detection and attention. Additionally, we propose the use of local attention, a variation of the standard MIL attention (Ilse et al., 2018).

## 2. Methods

In the MIL approach, we model voxels and axial slices in CTA scans as instances and bags of instances, respectively. We define a 3-class classification task on each slice to determine the affected hemisphere. The main steps are to calculate: 1) voxel class scores, 2) voxel attention weights and 3) the bag label during training and the infarct during inference.

**Dataset:** We included 456 acute ischemic stroke patients with early baseline CTA. The patients included in the test set had both baseline CTA and CTP scans. Follow-up infarct delineations are used to determine the class labels of the slices in baseline CTA scans. The train, validation, and test set contain 389, 5, and 32 scans, respectively. The scans are registered to a brain atlas.

**Model:** We use a $3D$ U-Net architecture similar to the one proposed by Nikolov et al. (2018) to calculate voxel class scores. To focus on the dis-similarity between hemispheres, we mirror the output of the U-Net and then calculate the difference between the original and the mirrored output. We refer to the resulting matrix as the *class score map*. The class score map is used as input to local attention (Figure 1) to calculate voxel weights. The main idea behind local attention is to apply a spatial constraint on the key instances and guide the network towards identifying the infarct voxels. We do this by dividing the brain into a left and a right hemisphere and calculating a class prediction for each hemisphere separately. Class predictions are calculated as a weighted average of voxel class scores. At inference, the infarct prediction is calculated by element-wise multiplying the attention weights and the class score map, and applying a threshold to create a binary prediction. Attention Layers in Figure 1 have the same architecture as proposed by Ilse et al. (2018).

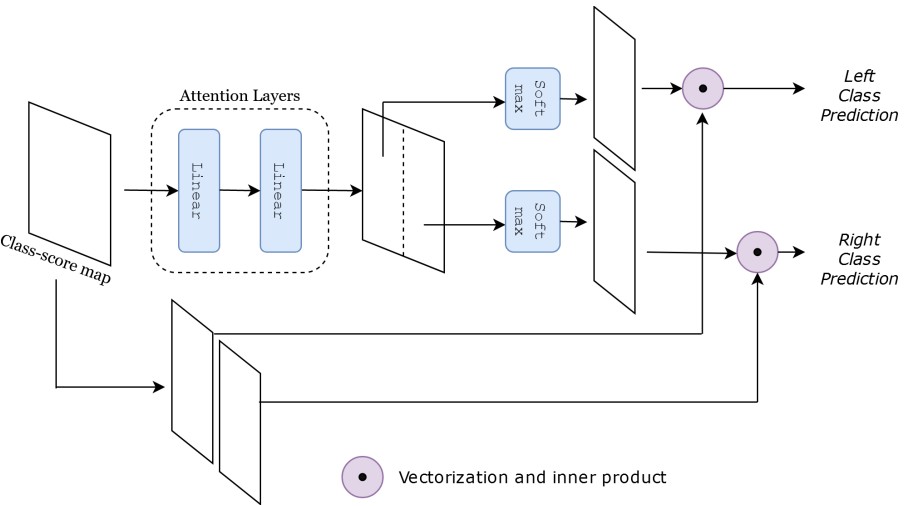

Figure 1: The local attention set-up.

**Experiments:** We compare the infarct volume predictions with 3 clinically validated CTP software packages: IntelliSpace Portal (ISP), syngo.via and RAPID. We use the intraclass correlation coefficient (ICC) for evaluating agreement. Since standard attention is designed

for binary classification tasks, this set-up is trained to predict the presence of infarct instead of the affected hemisphere.

## 3. Results and Discussion

The volumetric agreement of the infarct predictions with the CTP software presented in Table 1 shows superior performance of local attention over standard attention. Visual inspection of results shows that standard attention's key instances are always positioned in the same region and do not correlate with infarct location. Local attention, on the other hand, shows capability in detecting the location and extent of the infarct. This could be the result of dividing the brain into two hemispheres, which enforces a spatial constraint on local attention and results in a spatially meaningful placement of the key instances. Additionally, unlike standard attention, local attention can be applied on a multi-class classification task, and therefore has access to more information. Local attention predictions show fair agreement with RAPID and worse agreements with syngio.via and ISP. The level of agreement between the CTP software ranges between 0.23 and 0.58, indicating that the volumetric agreement of local attention with RAPID and syngio.via is comparable with the agreement among these clinically validated CTP software packages.

Table 1: Volumetric agreement of the MIL-based infarct ICC values.

| Attention Method | ICC with RAPID | ICC with syngio.via | ICC with ISP |
|---|---|---|---|
| Local attention | 0.4 | 0.2 | 0.025 |
| Standard attention | 0.084 | 0.022 | -0.052 |

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
