# OpenReview forum: "Multiple Instance Learning and Local Attention for Ischemic Stroke Infarct Segmentation on Early Acute Baseline CTA"
_MIDL.io/2021/Conference/Short — Submitted to MIDL 2021_

### Official Review · Reviewer_h388 · 2021-04-27

**Confidence:** 4
**Final Rating:** 2

**Summary:**

The authors present a weakly-supervised multiple instance learning approach for predicting the infarct core based on CTA images. The performance of the network is evaluated using standard vs local attention, and further compared to 3 clinically validated software packages that use CTP-based biomarkers. The results indicate that the proposed method with local attention produces comparable results to those obtained from the software packages.

**Strengths:**

1. The paper is very well written and is easy to read.
2. The clinical relevance of the study is clearly described.
3.  A relatively large dataset is available for this research.
4. Authors propose a novel methodological application.


**Weaknesses:**

1.	There is a lack of evaluation metrics. Authors should include other evaluation metrics for image segmentation, such as Dice similarity coefficient, to better describe and compare the segmentation accuracy. I would suggest adding a figure in the appendix that shows the segmentation results of a few patients.

2.	The train-test split is very skewed towards the training set. The results would be much more relevant if the test set were increased.

3.	The description of the dataset is critically missing. Where was the data acquired? What is the resolution of the input images? Were images pre-processed before being introduced to the model (i.e., skull stripped, temporally resampled)? How much time was there between baseline and follow-up images?

4.	The benefit of using MIL over other deep learning frameworks is not clear. It would be really helpful if the authors could give some advantages of MIL (i.e., memory requirements).

5.	In section 3, authors mention that *“The level of agreement between the CTP software ranges between 0.23 and 0.58…”*, but this is not supported by any reference or experiment. Showing results for the individual comparisons between software packages would help strengthen the information reported in Table 1.


**Deanonymize Review:**

no

**Justification Of The Rating:**

The topic of the research work is interesting. However, the current evaluation does not justify the significance of the proposed method and could lead to an erroneous assessment of the model performance.

**Paper Type:**

both

**Special Issue:**

no

---

### Official Review · Reviewer_9ijg · 2021-04-29

**Confidence:** 4
**Final Rating:** 3

**Summary:**

The paper propose to detect and localize infarcts in CT angiographs of the brain using a multiple instance learning method with a "local" attention mechanism. The method is developed and validated on CTs of 456 patients. The performance is compared to a similar method with "standard" attention mechanism, using three clinically validated software tools for infarct prediction from CT perfusion as reference. The "local" attention mechanism achieve much better agreement with the references than "standard" attention.

**Strengths:**

The main strengths are the idea of restricting attention to certain parts of the image and that the proposed local attention appears to clearly outperform the same model with standard attention.

I like that there is a comparison with clinically validated tools applied to CTP scans. Often when MIL is an appropriate choice there is a lack of instance predictions making evaluation of the localization and instance prediction very difficult.

**Weaknesses:**

There is something in the description of the model that confuse me. What do you actually optimize? Are your labels {no, left, right}? Can left and right co-occur? Otherwise I do not understand what you gain by splitting the output.

You mention that "Local attention [...] shows capability in detecting the location and extent of the infarct." but there are no numbers. In my view this would be the first metric to calculate and I think it should be included.

In Table 1 you present results on agreement with reference methods. But it is not clear what you compare. The table caption does not make sense to me: "Volumetric agreement of the MIL-based infarct ICC values." It must surely be agreement on some predictions, but which? Detection? Left/right localization? Something else? My guess is left/right localization, but it is just a guess.

Depending on what you measure I find the results a bit worrisome. No correlation with ISP is a big concern and ICC of 0.2 and 0.4 is also not a lot. You really need to comment on this.

**Deanonymize Review:**

no

**Detailed Comments:**

The class score map is the difference between the original and mirrored output of the U-Net. The two halves of this image are then each others mirror with opposite sign. Could you not then throw one side away?

**Justification Of The Rating:**

Although parts are unclear and I miss some details I think the paper is interesting because it considers how we can constrain attention mechanisms and it seems to work comparatively well. My main issue is that the results are not reported clearly, had they been I would consider a strong accept.

**Paper Type:**

both

**Special Issue:**

no

---

### Meta-Review · Program_Chairs · 2021-05-09

**Recommendation:** Reject
**Confidence:** 4

**Metareview:**

While the approach exploiting brain symmetry in attention guided segmentation is interesting and potentially useful, as reviewers point out, results are incomplete and seem quite bad (though better than without the proposed local attention). In my opinion it is too preliminary to accept for presentation at MIDL.

---

### Decision · Program_Chairs · 2021-05-11

Reject